# Long-term health impacts of COVID-19 among 242,712 adults in England

**Christina J. Atchison** [1,2], **Bethan Davies** [1,2,3], **Emily Cooper** [1], **Adam Lound**[1], **Matthew Whitaker**[1,3], **Adam Hampshire**[4], **Adriana Azor**[4], **Christl A. Donnelly** [1,5,6], **Marc Chadeau-Hyam** [1,7], **Graham S. Cooke** [2,8,9], **Helen Ward** [1,2,6,9,11] & **Paul Elliott** [1,2,3,7,9,10,11] ✉

The COVID-19 pandemic is having a lasting impact on health and well-being. We compare current self-reported health, quality of life and symptom profiles for people with ongoing symptoms following COVID-19 to those who have never tested positive for SARS-CoV-2 infection and those who have recovered from COVID-19. Overall, 276,840/800,000 (34·6%) of invited participants took part. Mental health and health-related quality of life were worse among participants with ongoing persistent symptoms post-COVID compared with those who had never had COVID-19 or had recovered. In this study, median duration of COVID-related symptoms (N = 130,251) was 1·3 weeks (inter-quartile range 6 days to 2 weeks), with 7·5% and 5·2% reporting ongoing symptoms ≥12 weeks and ≥52 weeks respectively. Female sex, ≥1 comorbidity and being infected when Wild-type variant was dominant were associated with higher probability of symptoms lasting ≥12 weeks and longer recovery time in those with persistent symptoms. Although COVID-19 is usually of short duration, some adults experience persistent and burdensome illness.

The UK has experienced one of the largest epidemics of COVID-19 (symptomatic SARS-CoV-2 infection) in Europe[1]. As well as the risk of hospitalisation and death from COVID-19 some people have a prolonged and debilitating illness that may continue for weeks or months (Long COVID or post-COVID syndrome)[2,3].

Estimates of symptom persistence following COVID-19 vary substantially, arguably due to heterogeneous study designs, study settings, follow-up periods and definitions. A recent meta-analysis of 194 studies with follow-up data from participants with confirmed or self-reported COVID-19 symptoms estimated a pooled prevalence of persistent symptoms at a mean follow-up of 126 days amongst hospitalised COVID-19 patients of 52.6% [95% CI 43.5, 61.6]. The most common persistent symptoms were fatigue, general pain or discomfort,

impaired sleep, breathlessness, and impaired usual activity[4]. In the non-hospitalised group, the pooled prevalence was 34.5% [95% CI 21.9, 49.7]. The most frequent persistent symptoms were fatigue, breathlessness, muscle pain or myalgia, affected sleep and loss of sense of smell[4]. Most existing studies are based on small sample size, unrepresentative study populations and low response rate, so estimates are unlikely to be representative of the general population[4].

Our own initial estimates from the REal-time Assessment of Community Transmission-2 (REACT-2) study, six rounds of repeat cross-sectional random samples of the population to evaluate community prevalence of SARS-CoV-2 anti-spike protein antibody positivity in England[5], suggested that 21.6% of adults with evidence of prior infection experienced one or more symptoms 12 weeks after their

[1]School of Public Health, Imperial College London, London, UK. [2]Imperial College Healthcare NHS Trust, London, UK. [3]MRC Centre for Environment and Health, Imperial College London, London, UK. [4]Department of Brain Sciences, Imperial College London, London, UK. [5]Department of Statistics, University of Oxford, Oxford, UK. [6]MRC Centre for Global infectious Disease Analysis and Abdul Latif Jameel Institute for Disease and Emergency Analytics, Imperial College London, London, UK. [7]Health Data Research (HDR) UK London at Imperial College, London, UK. [8]Department of Infectious Disease, Imperial College London, London, UK. [9]National Institute for Health Research Imperial Biomedical Research Centre, London, UK. [10]UK Dementia Research Institute at Imperial College London, London, UK. [11]These authors jointly supervised this work: Helen Ward, Paul Elliott. ✉e-mail: p.elliott@imperial.ac.uk

initial illness[6]. However, this study lacked a negative control group. Recently, the Long-COVID in Scotland study (Long-CISS), a nationwide study including people with severe, mild and asymptomatic infections and a never-infected comparison group, found that 8% of symptomatic participants had not recovered by 6 or 12 months[7]. Participants with previous symptomatic infection were more likely to self-report 24 (of the 26 surveyed) persistent symptoms than people never infected. The largest effect sizes were observed for changes in taste and smell, breathlessness, chest pain, palpitations, and confusion. Similarly, a nationwide population cohort study in the Netherlands (Lifelines COVID-19) with COVID-19-positive cases and matched negative controls concluded that core symptoms of Long COVID were chest pain, difficulties with breathing, lump in throat, pain when breathing, painful muscles, heavy arms or legs, ageusia or anosmia, feeling hot and cold alternately, tingling extremities, and general tiredness[8].

The REACT programme is one of the world's largest and most comprehensive coronavirus monitoring studies. In addition to REACT-2 described above, the REACT-1 study included 19 rounds of cross-sectional random samples of the population to track community SARS-CoV-2 infection with PCR tests[9]. The REACT programme is able to identify individuals with persistent symptoms who have not been hospitalised and to compare them with people whose symptoms have resolved and those who have never had COVID-19. In this study, we use data from a follow-up survey of REACT participants to describe the duration of symptoms in people with a history of symptomatic infection, assess factors associated with symptom persistence beyond 12 weeks and with recovery after that point. We also compare current self-reported health and quality of life and specific symptoms for those with persistent symptoms to those who have never had COVID-19 or have recovered.

## Results

### Overview of study population
Of the 800,000 REACT participants who were sent invitations between 1 August–1 December 2022, 282,780 (35·3%) registered for the study, of whom 276,840 (97·9%) completed the questionnaire. Differential non-response was observed by sociodemographic characteristics, including age, sex, ethnicity, and deprivation (Supplementary Table 1).

Of those who completed the questionnaire, 266,854/276,840 (96·4%) reported whether they had a confirmed SARS-CoV-2 infection. In total, 157,668/266,854 (59·1%) participants had tested positive for SARS-CoV-2. Overall, 24,142 respondents were excluded because COVID-19 episode date was within 12 weeks of their survey completion date (Supplementary Fig. 1). Table 1 shows the key sociodemographic and COVID-19 characteristics of all participants included in the study.

### Factors associated with persistent symptoms following COVID-19 and recovery
Of the 133,526 people who reported at least one episode of test-confirmed SARS-CoV-2 infection, 3275 reported no symptoms, and 130,251 reported symptomatic COVID-19. In these 130,251 participants, the median duration of reported symptoms was 1.3 weeks, (mean 5·4 weeks, range 1 day to 3·0 years, IQR 6 days to 2 weeks), with 10·2%, 7·5%, and 5·2% reporting ongoing symptoms beyond 4 weeks, 12 weeks and 52 weeks, respectively.

Figure 1 shows the Kaplan Meier Survival Curve of time to symptom end date in those self-reporting symptomatic SARS-CoV-2 infection, overall and by dominant variant at the time of infection. Figure 2 shows factors associated with having symptoms following COVID-19 persisting for ≥12 weeks ("Long COVID", LC) and ≥52 weeks ("very Long COVID", VLC) compared to those who were asymptomatic or whose symptoms resolved within 4 weeks (Supplementary Table 2). In mutually adjusted logistic regression models, LC and VLC were both associated with being female compared to male (adjusted odds ratio [aOR] 1·42 [95% CI 1·35, 1·50] and 1·49 [1·38, 1·62] for LC and VLC respectively), having ≥1

comorbidities compared to no comorbidities (aOR 1·31 [1·19, 1·44], 1·52 [1·31, 1·76] respectively for 1, and 1·46 [1·27, 1·75], 2·35 [1·85, 2·97] for ≥2 comorbidities), and having had moderate or severe symptoms compared to mild at the time of infection (aOR 1·76 [1·63, 1·89], 1·47 [1·32, 1·64] respectively for moderate and 4·87 [4·52, 5·25], 3·55 [3·19, 3·96] for severe). The odds of LC and VLC were lower in people of Asian than white ethnicity (aOR 0·80 [0·69, 0·93], 0·71 [0·57, 0·88]), and in people infected at a time when Alpha (aOR 0·60 [0·56, 0·64], 0·59 [0·54, 0·64]), Delta (OR 0·38 [0·35, 0·41], 0·32 [0·29, 0·36]) and Omicron (OR 0·12 [0·11, 0·13] for LC, insufficient follow-up time for VLC) were dominant compared to Wild-type. In a sensitivity analysis to test whether the lower risk with more recent variants was due to unmeasured time-varying factors we restricted the comparison to cases close to the transition period and also found a reduction in risk for Alpha compared to Wild-type and Omicron compared to Delta (Supplementary Table 3).

There was also a gradient of reducing odds with lower deprivation (Fig. 2). There was a suggestion of lower risk of LC, but not VLC, in older compared to younger people (Fig. 2).

We fitted mutually adjusted Accelerated Failure Time models to assess factors associated with the rate of recovery from persistent symptoms (Supplementary Table 4). For people who had LC (≥12 weeks), longer time to recovery was found for females (adjusted time to recovery [aTR] 1·14 [1·06, 1·23] compared to males) and people with comorbidities (aTR 1·24 [1·08, 1·42] and 2·05 [1·58, 2·66] respectively for 1 and ≥2 comorbidities compared to no comorbidities). Shorter time to recovery of LC was found in people of other and mixed ethnicity (aTR 0·63 [0·45, 0·89] and 0·75 [0·57, 0·99] for other and mixed compared to white), people living in the least deprived areas (aTR 0·78 [0·69, 0·88] compared to most deprived), not-current smokers (aTR 0·73 [0·62, 0·86] compared to current smokers), later variants at the time of infection (aTR 0·79 [0·72, 0·86], 0·89 [0·79, 0·99], 0·69 [0·61, 0·78] for Alpha, Delta and Omicron respectively compared to Wild-type).

### Current symptom profile and health-related quality of life
In our study population, the most common symptoms experienced by individuals with ongoing persistent symptoms were mild fatigue (66·9%), difficulty thinking or concentrating (54·9%) and joint pains (54·6%). However, mild fatigue and joint pains were also common in those with no history of COVID-19 and in those in whom symptoms had resolved (Fig. 3). The greatest difference in symptom prevalence between those with ongoing persistent symptoms and other participants were for loss or change of sense of smell (aOR 9·31, [8·64, 10·04]) or taste (aOR 8·47; [7·85, 9·15]), shortness of breath (aOR 6·69; [6·29, 7·12]), severe fatigue (aOR 6·19; [5·66, 6·77]), difficulty thinking or concentrating (aOR 4·97; [4·68, 5·27]), chest tightness or pain (aOR 4·71; [4·37, 5·08]) and poor memory (aOR 4·40; [4·15, 4·66]) (Fig. 3).

Participants with ongoing symptoms lasting ≥12 weeks following COVID-19 reported worse current health, including a higher number of symptoms within the last two weeks, and greater reduction in ability to carry out daily activities due to these symptoms (Table 2). Post-exertional malaise, characterised by asking respondents who reported fatigue about worsening of fatigue symptoms after minimal physical and mental effort, and whether exercise makes fatigue symptoms worse, was also more common in individuals reporting ongoing persistent symptoms (Table 2, Supplementary Table 7). Additionally, worse mental health and health-related quality of life were reported by participants with ongoing symptoms lasting ≥12 weeks (Table 2, Supplementary Table 7). Table 2 also shows that for the 3221 people who previously had post-COVID-19 symptoms lasting ≥12 weeks and report having recovered, their health status is broadly similar to those with shorter recovery or who never had COVID-19.

## Discussion
We show that symptomatic SARS-CoV-2 infection in England in adults is usually short-lived with most people reporting a short illness with

**Table 1 | Comparison of participant sociodemographic and COVID-19 characteristics by COVID-19 history (n = 242,712)[1]**

| | N (%) | No COVID No. (%) | Asymptomatic or resolved short COVID < 4 weeks No. (%) | Resolved short COVID ≥ 4 to < 12 weeks No. (%) | Resolved persistent COVID | | Ongoing persistent COVID | |
|---|---|---|---|---|---|---|---|---|
| | | | | | ≥12 to < 52 weeks No. (%) | ≥52 weeks No. (%) | ≥12 to < 52 weeks No. (%) | ≥52 weeks No. (%) |
| | 242,712 | 109,186 (45·0) | 117,022 (48·2) | 7510 (3·1) | 2,323 (0·96) | 898 (0·37) | 2551 (1·1) | 3222 (1·3) |
| **Age*** | | | | | | | | |
| 18 to 24 | 6747 (2·8) | 2311 (34·3) | 3887 (57·6) | 199 (3·0) | 90 (1·3) | 43 (0·64) | 84 (1·2) | 133 (2·0) |
| 25 to 34 | 16,040 (6·6) | 4811 (30·0) | 9831 (61·3) | 590 (3·7) | 195 (1·2) | 64 (0·40) | 279 (1·7) | 270 (1.7) |
| 35 to 44 | 27,857 (11·5) | 7965 (28·6) | 16,971 (60·9) | 1272 (4·6) | 402 (1·4) | 140 (0·50) | 536 (1·9) | 571 (2.1) |
| 45 to 54 | 42,264 (17·4) | 14,936 (35·3) | 23,319 (55·2) | 1779 (4·2) | 595 (1·4) | 180 (0·43) | 607 (1·4) | 848 (2.0) |
| 55 to 64 | 62,698 (25·8) | 28,181 (45·0) | 30,139 (48·1) | 2023 (3·2) | 636 (1·0) | 249 (0·40) | 585 (0·93) | 885 (1.4) |
| 65 to 74 | 60,411 (24·9) | 33,063 (54·7) | 24,828 (41·1) | 1290 (2·1) | 319 (0·53) | 172 (0·28) | 334 (0·55) | 405 (0.67) |
| 75+ | 26,695 (11·0) | 17,919 (67·1) | 8047 (30·1) | 357 (1·3) | 86 (0·32) | 50 (0·19) | 126 (0·47) | 110 (0.41) |
| **Sex at birth*** | | | | | | | | |
| Male | 100,898 (41·6) | 48,308 (47·9) | 47,368 (47·0) | 2473 (2·5) | 760 (0·75) | 332 (0·33) | 702 (0·70) | 955 (0.95) |
| Female | 141,807 (58·4) | 60,874 (42·9) | 69,651 (49·1) | 537 (3·6) | 1563 (1·1) | 566 (0·40) | 1849 (1·3) | 2267 (1.6) |
| **Ethnicity*** | | | | | | | | |
| White | 227,112 (94·6) | 102,265 (45·0) | 109,600 (48·3) | 6895 (3·0) | 2137 (0·94) | 829 (0·37) | 2387 (1·1) | 2999 (1.3) |
| Mixed | 2775 (1·2) | 1007 (36·3) | 1507 (54·3) | 118 (4·3) | 47 (1·7) | 10 (0·36) | 45 (1·6) | 41 (1.5) |
| Asian | 6435 (2·7) | 2840 (44·1) | 3085 (47·9) | 260 (4·0) | 70 (1·1) | 28 (0·44) | 65 (1·0) | 87 (1.4) |
| Black | 1987 (0·83) | 985 (49·6) | 846 (42·6) | 79 (4·0) | 22 (1·1) | 12 (0·60) | 14 (0·70) | 29 (1.5) |
| Other | 1697 (0·71) | 692 (40·8) | 833 (49·1) | 87 (5·1) | 30 (1·8) | 9 (0·53) | 20 (1·2) | 26 (1.5) |
| **IMD*** | | | | | | | | |
| 1 – most deprived | 19,888 (8·4) | 8604 (43·3) | 9473 (47·6) | 744 (3·7) | 227 (1·1) | 95 (0·48) | 276 (1·4) | 469 (2.4) |
| 2 | 35,564 (15·0) | 15,847 (44·6) | 17,063 (48·0) | 1146 (3·2) | 369 (1·0) | 130 (0·37) | 429 (1·2) | 580 (1.6) |
| 3 | 50,627 (21·4) | 23,183 (45·8) | 24,032 (47·5) | 1547 (3·1) | 500 (0·99) | 196 (0·39) | 509 (1·0) | 660 (1.3) |
| 4 | 60,165 (25·4) | 27,187 (45·2) | 29,103 (48·4) | 1794 (3·0) | 531 (0·88) | 197 (0·33) | 617 (1·0) | 736 (1.2) |
| 5 – least deprived | 70,419 (29·8) | 31,722 (45·1) | 34,294 (48·7) | 2115 (3·0) | 652 (0·93) | 254 (0·36) | 668 (0·95) | 714 (1.0) |
| **Comorbidities*** | | | | | | | | |
| 0 | 156,433 (64·5) | 34,593 (22·1) | 106,790 (68·3) | 6920 (4·4) | 2153 (1·4) | 843 (0·54) | 2254 (1·4) | 2880 (1.8) |
| 1 | 38,811 (16·0) | 30,347 (78·2) | 7402 (19·1) | 425 (1·1) | 143 (0·37) | 41 (0·11) | 220 (0·57) | 233 (0.60) |
| 2 or more | 47,468 (19·6) | 44,246 (93·2) | 2830 (6·0) | 165 (0·35) | 27 (0·06) | 14 (0·03) | 77 (0·16) | 109 (0.23) |
| **Smoking Status*** | | | | | | | | |
| Current smoker | 13,257 (5.7) | 7096 (53.5) | 5365 (40.5) | 333 (2.5) | 85 (0.64) | 31 (0.23) | 165 (1.2) | 182 (1.4) |
| Not current smoker | 217,796 (94.3) | 98,104 (45.0) | 105,037 (48.2) | 6795 (3.1) | 2080 (0.96) | 785 (0.36) | 2200 (1.0) | 2795 (1.3) |
| **Severity of initial SARS-CoV-2 infection*** | | | | | | | | |
| No symptoms | 6206 (4·7) | - | 6206 (100·0) | 0 (0·0) | 0 (0·0) | 0 (0·0) | 0 (0·0) | 0 (0·0) |
| Mild symptoms | 36,207 (27·2) | - | 34,346 (94·9) | 724 (2·0) | 228 (0·63) | 230 (0·64) | 345 (0·95) | 334 (0·92) |
| Moderate symptoms | 64,958 (48·8) | - | 58,023 (89·3) | 3308 (5·1) | 916 (1·4) | 404 (0·62) | 1174 (1·8) | 1133 (1·7) |
| Severe symptoms | 25,682 (19·3) | - | 17,974 (70·0) | 3478 (13·5) | 1179 (4·6) | 264 (1·0) | 1032 (4·0) | 1755 (6·8) |
| **Dominant strain at symptom onset or positive test*** | | | | | | | | |
| Wild type (before Dec 2020) | 15,174 (11·6) | - | 10,157 (66·9) | 1924 (12·7) | 908 (6·0) | 542 (3·6) | 0 (0·0) | 1,643 (10·8) |
| Alpha (Dec 2020-April 2021) | 12,054 (9·2) | - | 9124 (75·7) | 1264 (10·5) | 508 (4·2) | 260 (2·2) | 0 (0·0) | 898 (7·4) |
| Delta (May 2021-mid-Dec 2021) | 19,993 (15·3) | - | 16,679 (83·4) | 1464 (7·3) | 462 (2·3) | 96 (0·48) | 611 (3·1) | 681 (3·4) |
| Omicron (after mid-Dec 2021) | 83,544 (63·9) | - | 78,301 (93·7) | 2858 (3·4) | 445 (0·53) | 0 (0·0) | 1940 (2·3) | 0 (0·0) |
| **Vaccination status at symptom onset or positive test*** | | | | | | | | |
| 0 | 67,013 (50·2) | - | 56,079 (83·7) | 4653 (6·9) | 1759 (2·6) | 812 (1·2) | 844 (1·3) | 2866 (4·3) |
| 1 | 2147 (1·6) | - | 1804 (84·0) | 125 (5·8) | 44 (2·1) | 39 (1·8) | 24 (1·1) | 111 (5·2) |
| 2 or more | 64,366 (48·2) | - | 59,139 (91·9) | 2732 (4·2) | 520 (0·81) | 47 (0·07) | 1683 (2·6) | 245 (0·38) |

Unweighted survey data presented. 1. Percentages are calculated by category after the exclusion of missing data for that variable with the denominator being all participants in the study, including those with no history of SARS-CoV-2 infection; p values show the association of COVID-19 status with sociodemographic or COVID-19 characteristic. A χ² test (two-sided) was used to identify differences in proportions across COVID-19 categories. No adjustments were made for multiple comparisons. *P < 0·0001 (exact P-values: age $P = 0.0000002$, sex $P = 0.0000004$, ethnicity $P = 0.00006$, IMD $P = 0.0000005$, comorbidities $P = 0.0000001$, smoking status $P = 0.00004$, severity of initial infection $P = 0.0000001$, dominant variant at time of infection $P = 0.0000002$, vaccination status $P = 0.000003$)

symptom resolution within 2 weeks. However, in our study population, one in 10 people with symptomatic SARS-CoV-2 infection report symptoms for more than 4 weeks, one in 13 for more than 12 weeks (meeting the WHO definition for "post COVID-19 condition (Long COVID)"[10]), and 1 in 20 for more than 52 weeks. In our study population, 69% of those with persistent symptoms at 12 weeks still had symptoms at 52 weeks, meaning that 31% recovered within a year. We found that female sex, higher deprivation, having a pre-existing health condition, more severe symptoms at onset, and being infected when the original Wild-type variant was dominant was associated with

having symptoms persisting for ≥12 weeks and ≥52 weeks. The above variables have previously been identified as risk factors for Long COVID[2,7,11–13]. We found a suggestion of lower reporting of persistent symptoms in older ages unlike another population-based study in the UK which found a positive linear association between age and Long COVID[7].

The variant at the time of infection, initial severity and presence of pre-existing health conditions had the biggest impact on persistent symptoms, consistent with previous findings[6,7,14,15]. Compared to Wild-type, those infected when Omicron was dominant were 88% less likely to report symptoms beyond 12 weeks; this may reflect changing immunity in the population from previous exposure to the virus and vaccination. A recent case-control study conducted in the UK found lower odds of Long COVID with the Omicron versus the Delta variant, ranging from OR 0·2 (95% CI 0·2, 0·3) in those vaccinated >6 months prior to infection to 0·5 (95% CI 0·4, 0·6) in those vaccinated <3 months prior to infection[16]. We did not find conclusive evidence of effectiveness of vaccination against Long COVID. Vaccination reduces the severity of COVID-19[17] and it may be through this indirect route that it has an impact on the risk of persistent symptoms post-infection. However, recent systematic reviews suggest that vaccination before SARS-CoV-2 infection could reduce the risk of subsequent Long COVID[13,18,19].

The reporting of current symptoms was high across all groups in our study. For example, while 66·9%, 54·9% and 54·6% of individuals with ongoing persistent symptoms post-COVID-19 reported currently experiencing mild fatigue, difficulty thinking or concentrating and joint pains, respectively, the prevalence of these symptoms in those who never had COVID-19 was also high, at 31·1%, 15·2% and 35·5%. Indeed, a high level of symptom reporting was also observed for those who had recovered from COVID-19; the prevalence of mild fatigue, difficulty thinking or concentrating and joint pains in those with asymptomatic SARS-CoV-2 infection or who had recovered from COVID-19 within 4 weeks was 38·3%, 21·3% and 34·4%, respectively. These findings of high symptom prevalence in comparison groups have been observed elsewhere[7,20,21], and could be due to higher participation in studies of people with current symptoms. Alternatively, this may reflect the timing of our survey which included months with high levels of upper-respiratory and influenza-like illness in the population. However, our data did show that the most specific persistent symptoms following COVID-19 were loss or change of sense of smell or taste, shortness of breath, severe fatigue, and difficulty thinking or concentrating, which were nine, seven, six and five times more likely, respectively, than in other participants. Of the few studies with a COVID-19-negative comparator group, one showed that COVID-19 cases had a higher likelihood of mood disorder, anxiety, and insomnia when compared to people with influenza or respiratory tract infection[22]. Another study found that in comparison to community controls, COVID-19 cases had a higher prevalence of symptoms at 6- and 9-months, including fatigue, sleep difficulties, hair loss, smell disorder, taste disorder, palpitations, chest pain, and headaches[23].

There were substantial differences in currently reported health and well-being between individuals reporting ongoing persistent post-COVID-19 symptoms and those who had never had COVID-19 or had recovered, consistent with published evidence[7,14,24]. Encouragingly, those whose symptoms had resolved, even after 52 weeks, had general health and quality of life scores similar to those with no COVID-19 history or who recovered quickly. The dyspnoea and post-exertional malaise (PEM) scales were asked of everyone reporting shortness of breath (Dyspnoea 12) or fatigue, and individuals reporting persistent symptoms following COVID-19 scored higher (i.e. worse symptoms) than others, suggesting these symptoms may be more

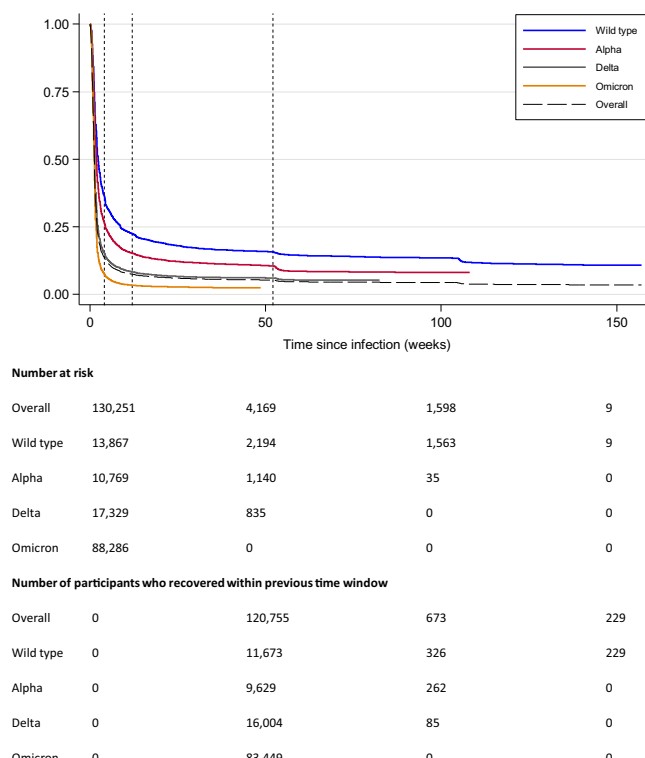

**Number at risk**

| | | | | |
|---|---|---|---|---|
| Overall | 130,251 | 4,169 | 1,598 | 9 |
| Wild type | 13,867 | 2,194 | 1,563 | 9 |
| Alpha | 10,769 | 1,140 | 35 | 0 |
| Delta | 17,329 | 835 | 0 | 0 |
| Omicron | 88,286 | 0 | 0 | 0 |

**Number of participants who recovered within previous time window**

| | | | | |
|---|---|---|---|---|
| Overall | 0 | 120,755 | 673 | 229 |
| Wild type | 0 | 11,673 | 326 | 229 |
| Alpha | 0 | 9,629 | 262 | 0 |
| Delta | 0 | 16,004 | 85 | 0 |
| Omicron | 0 | 83,449 | 0 | 0 |

**Fig. 1 | Kaplan–Meier survival curve of time to symptom end date in those self-reporting symptomatic SARS-CoV-2 infection, overall and by dominant variant at the time of infection.** The curve shows the probability that a participant continues to have symptoms beyond time *t*. Participants infected at a time when Wild-type (blue) was dominant had a higher probability of symptoms continuing beyond time t compared to Alpha (red), Delta (grey) and Omicron (orange). The number at risk table below the curve shows, overall and by dominant variant at the time of infection, the number at risk at any specific time point. This is equal to the total number of participants remaining in the study including any individuals who experience the event of interest (symptom end date) or participants who are censored at this time point. The unit of time is "weeks," so the number at risk is those participants who have not yet experienced the event of interest or been censored at the beginning of the week (before any event or censoring could occur). The number of participants who recovered within the previous time window are shown below the number at risk table. This is equal to the total number of participants with a symptom end date within the time window ending at this time point.

specific. A meta-analysis of 12 studies that evaluated health-related quality of life in individuals with Long COVID reported a pooled prevalence of poor quality of life (EQ5D Visual Analogue Scale - EQ-VAS) of 59% (95% CI 42, 75)[24]. Similarly, the Long-COVID in Scotland study found that symptomatic SARS-CoV-2 infection was associated with a wide range of impaired daily activities and reduced health-related quality of life[7].

A strength of our study is that we have addressed some of the limitations of existing studies by having a comparison group and including people in the general population who had severe, mild, and asymptomatic SARS-CoV-2 infections. We compared contemporaneous symptom profiles of community-based adults reporting ongoing persistent symptoms post-COVID-19 versus those who have never had COVID-19 or have recovered. Our study is the largest yet to look at these questions and goes further than previous questionnaire-based studies with COVID-19 negative[8] and never-infected[7] controls by identifying factors associated not only with recovery (yes/no) but rate of recovery. Indeed, our findings highlight the importance of having a comparator cohort of participants who tested negative and experienced the pandemic and national lockdowns. However, we acknowledge the possibility of misclassification

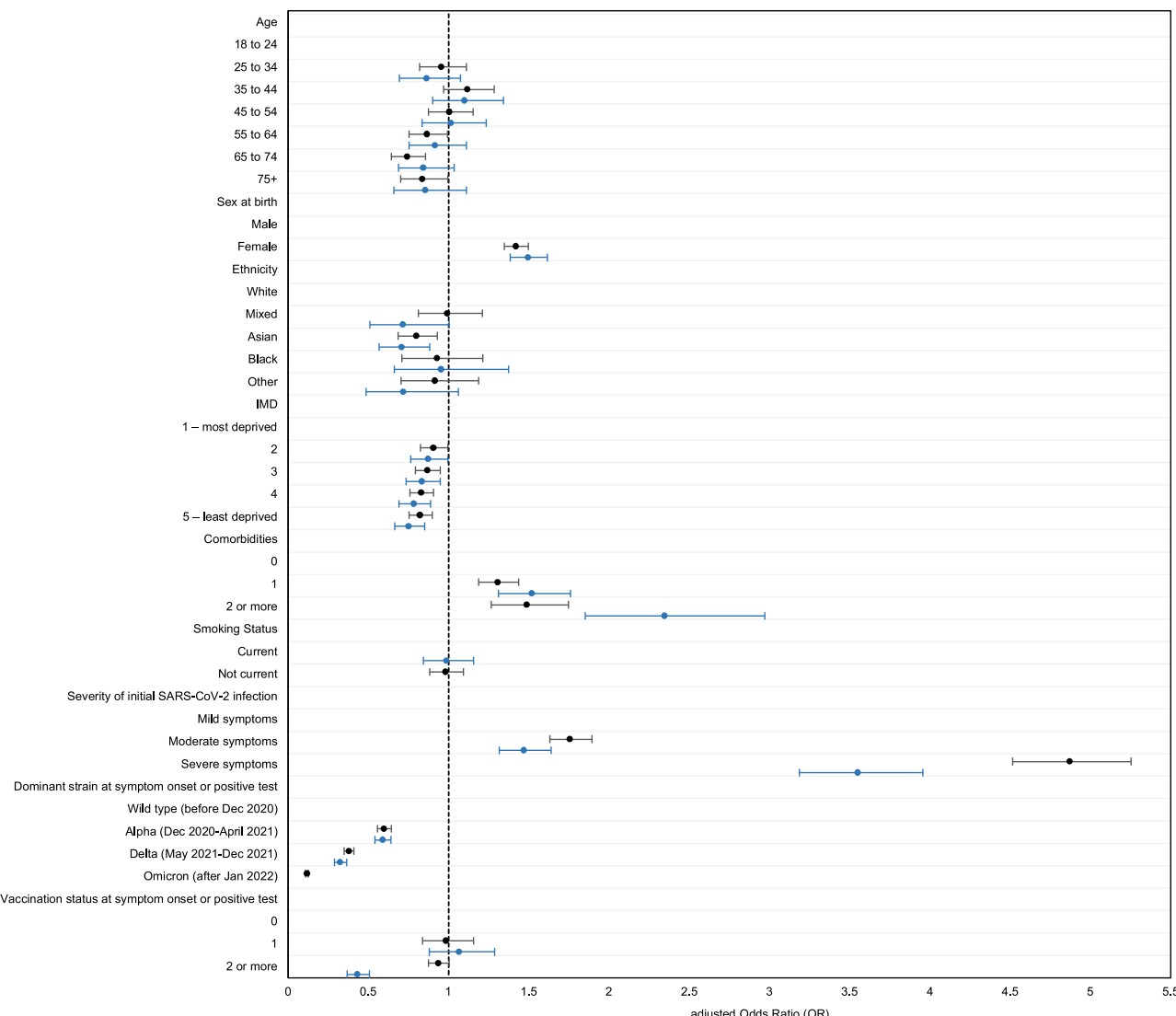

**Fig. 2 | Factors associated with persistent symptoms following COVID-19 lasting i) ≥12 weeks and ii) ≥52 weeks versus those who reported being asymptomatic or symptoms resolved within 4 weeks.** Logistic regression models with one or more COVID-19 symptoms lasting ≥12 weeks (y/n) or ≥52 weeks (y/n) as the binary outcome variables. Modelling of persistent symptoms as a function of biological and demographic variables. In the forest plot, data were presented as adjusted odds ratios (central dot) and 95% confidence intervals (bars). Adjusted odds ratios compare participants with persistent symptoms lasting i) ≥12 weeks (black) or ii) ≥52 weeks (blue) with those who reported being asymptomatic or symptoms resolved within 4 weeks (n = 126,016 participants for ≥12 weeks analysis / n = 121,142 participants for ≥52 weeks analysis). Mutually adjusted for age, sex, ethnicity, IMD, comorbidities, smoking status, severity of initial infection, dominant variant at the time of infection, and vaccination status. Data used: Supplementary Table 2.

bias in our comparator groups as infections may have gone undetected particularly in stages of the pandemic when free universal testing was not available in the UK.

We also recognise that the subjective nature of symptoms creates the potential for reporting and recall bias. We used information regarding presence and duration of symptoms rather than whether participants described themselves as having "Long COVID" to reduce potential reporting bias. The data on symptoms at the time of PCR testing were retrospective which introduces the possibility of recall bias, although we have previously shown that REACT participant reports of symptom onset date closely mirrored the epidemic curve[25]. There is also a risk that recall bias may have differentially affected reporting of symptoms by participants infected at different times, along with other time-varying factors, such as behaviour, seasonal weather patterns and changing pandemic restrictions, knowledge and expectations[26], which may account for at least part of the association between persistent symptoms and Wild-type infection. However, studies looking at individuals with confirmed infections of different SARS-CoV-2 strains also show lower risks with more recent variants[27,28].

We used validated instruments to assess mental health[29,30], quality of life[31]. dyspnoea[32], and fatigue[33] but recognise the limitations of self-reporting and floor and ceiling effects (i.e, if a higher percentage of individuals achieve either maximum or minimum scores). The PHQ-9 scale used is a diagnostic tool for depression. However, some of the somatic questions have been found to be strongly correlated with symptoms that are common in Long COVID, including fatigue, sleep disruption and brain fog[34]. As such, by using this scale we might be overestimating the level of depression. This issue was raised by Re'em et al. who suggest the PHQ-2[35] screening criteria may be more appropriate for Long COVID as they do not include somatic items and simply require a score of 3 or more from the first two questions of PHQ-9[34]. The percentage of participants in our study across all COVID-19

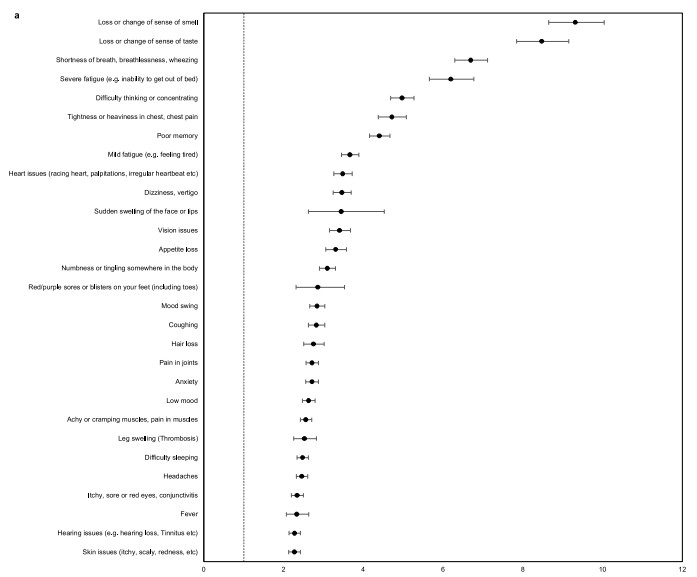
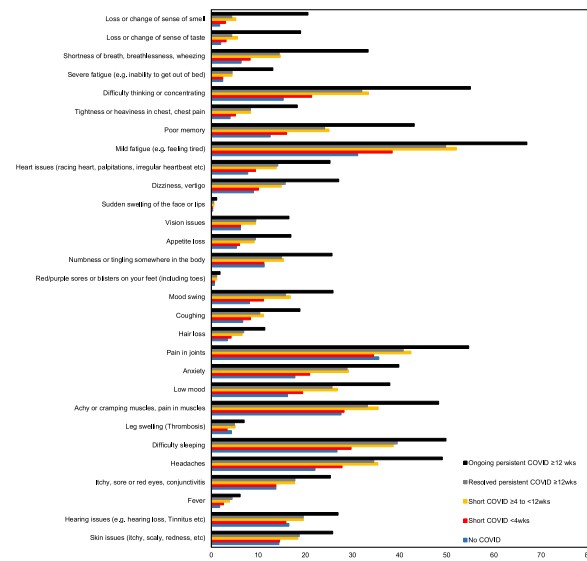

**Fig. 3 | Current symptoms profile by COVID-19 history a)** Forest plot of current symptoms in those reporting ongoing persistent symptoms following COVID-19 versus all other respondents, and **b)** Prevalence of current symptoms by COVID-19 history. Panel a shows the results of logistic regression models with 29 individual symptoms currently experienced (y/n) as the binary outcome variable and COVID-19 history as the primary exposure variable of interest (*n* = 242,712 participants). In the forest plot, data were presented as adjusted odds ratios (central dot) and 95% confidence intervals (bars). Odds ratios adjusted for age, sex, ethnicity, IMD, comorbidities, smoking status. Panel b shows the prevalence of 29 symptoms surveyed (*n* = 242,712 participants). Unweighted survey data presented. Data used: Supplementary Tables 5, 6.

categories with a PHQ-2 score ≥3 was lower than the percentage with PHQ-9 ≥10 and the difference was more marked in those with ongoing persistent symptoms post-COVID-19 (Supplementary Table 7, Supplementary Table 8).

Our questionnaire response rate was 34·6%; response rates in the Long-COVID in Scotland Study and a Dutch population-based cohort were 16% and 33-39%, respectively[7,8]. Like these studies, our participants were more likely to be female, older, of white ethnicity and from the least deprived areas compared with the general adult population. These issues might cause selection bias in our study; however, we did not observe substantial differences between those invited and those who participated in the study on the measured sociodemographic characteristics (Supplementary Table 1).

A further limitation is that we do not present estimates for the population prevalence of persistent symptoms. To do so would require weighting but production of weights is far from straightforward given the composition of our sample. The probability of being in the sample was dependent upon the composition of the base population, varying response rates by sociodemographic group and across REACT-1 and REACT-2 rounds. We also oversampled participants who tested positive for SARS-CoV-2 and who reported persistent symptoms. Producing weights that take account of all these factors would involve making extensive assumptions which would likely introduce unknown biases.

In summary, our study provides timely data about the effect and implications of the pandemic on adults in England with and without ongoing persistent symptoms following COVID-19. Although COVID-19 is usually of short duration, some adults experience persistent and burdensome illness, although a sizeable proportion still recover after a prolonged period. Differences in symptoms and recovery are being further explored in an in-depth interview study with REACT participants to further understand the varying presentations and trajectories of post-COVID conditions[36].

## Methods
### Study sample frame: the REACT programme
The REACT programme sampled random cross-sections of the population in England to quantify community prevalence of virus by RT-PCR

(REACT-1, 19 rounds between May 2020 and March 2022) and of IgG anti-SARS-CoV-2 antibody based on a self-administered lateral flow immunoassay (LFIA) test (REACT-2, six rounds between June 2020 and May 2021)[37]. Methods for the studies are published elsewhere[5,9,37]. The average response rate across both studies and all rounds was 23·4%. Each round of data collection included between 90,000 and 210,000 participants[38]. Overall, 3,099,386 adults registered to take part in the REACT programme. Of these participants, 2,494,309/3,099,386 (80·5%) consented to both recontact and data linkage of routine health records.

### Study design and participants
In this study, we aimed for a sample size of at least 160,000. Assuming a 20·0% response rate, we obtained a sample of 800,000 adults aged ≥18 years using as a sample frame REACT-1 and REACT-2 participants who had consented to both re-contact and data linkage (*n* = 2,494,309) (Supplementary Fig. 1). Personalised invitations were sent via email for one round of data collection between 1 August–1 December 2022. To increase our sample of individuals with persistent symptoms following COVID-19 we first invited all individuals in the following subgroups:

1. Individuals from REACT-1 or REACT-2 with a previous history of self-reported test confirmed or suspected COVID-19 who reported persistent symptoms of ≥12 weeks (*n* = 52,501)
2. Individuals from REACT-1 who tested positive for SARS-CoV-2 as part of the study (*n* = 13,482)
3. Individuals from REACT-2 who tested positive for SARS-CoV-2 IgG as part of the study and had not been vaccinated at the time (*n* = 85,757)

To achieve the 800,000-participant size, a random sample (*n* = 648,260) of all remaining adults not meeting the above criteria was selected.

Participants registered via an online portal. Those registered completed an online questionnaire[38]. It was designed to collect data on current health, well-being, functionality and recent symptoms, followed by questions about history of COVID-19 (SARS-CoV-2 PCR and

**Table 2 | Current symptom profile and health-related quality of life of participants by COVID-19 history (n = 242,712)**

| | No COVID No. (%) | Asymptomatic or resolved short COVID < 4 weeks No. (%) | Resolved short COVID ≥ 4 to < 12 weeks No. (%) | Resolved persistent COVID | | Ongoing persistent COVID | |
| | | | | ≥12 to < 52 weeks No. (%) | ≥52 weeks No. (%) | ≥12 to < 52 weeks No. (%) | ≥52 weeks No. (%) |
|---|---|---|---|---|---|---|---|
| N = 242,712 | 109,186 (45·0) | 117,022 (48·2) | 7510 (3·1) | 2323 (0·96) | 898 (0·37) | 2551 (1·1) | 3222 (1·3) |
| **Health Status*** | | | | | | | |
| Good | 83,200 (76·3) | 94,643 (80·9) | 5386 (71·8) | 1719 (74·1) | 675 (75·3) | 1421 (55·7) | 1575 (49·0) |
| Fair | 21,337 (19·6) | 19,146 (16·4) | 1780 (23·7) | 514 (22·2) | 184 (20·5) | 864 (33·9) | 1204 (37·4) |
| Bad | 4537 (4·2) | 3161 (2·7) | 341 (4·5) | 86 (3·7) | 37 (4·1) | 265 (10·4) | 438 (10·4) |
| **No. of current symptoms*** | | | | | | | |
| Median (IQR) | 2·0 (0, 5) | 3·0 (1, 6) | 4·0 (2, 8) | 4·0 (2, 8) | 4·0 (1, 7) | 7·0 (4, 11) | 8·0 (5, 12) |
| **No. of symptoms at infection*** | | | | | | | |
| Median (IQR) | ·· | 7·0 (4, 11) | 11·0 (8, 15) | 11·0 (7, 15) | 9·0 (5, 13) | 11·0 (7, 16) | 12·0 (8, 16) |
| **Duration of COVID-19 symptoms* (weeks)** | | | | | | | |
| Median (IQR) | ·· | 1·1 (0·71, 1·6) | 5·7 (4·4, 7·6) | 18·4 (14·0, 26·1) | 56·6 (53·4, 104·7) | 31·0 (21·4, 41·0) | 96·0 (80·9, 121·7) |
| Range | ·· | 0·0, 3·9 | 4·0, 11·9 | 12·0, 51·9 | 52·0, 142·4 | 12·0, 51·9 | 52·0, 157·0 |
| **Reduction in daily activities*** | | | | | | | |
| A lot | 10,115 (12·2) | 8,959 (9·6) | 875 (13·2) | 294 (14·4) | 104 (13·3) | 650 (26·7) | 984 (31·7) |
| A little | 37,430 (45·0) | 45,344 (48·8) | 3762 (56·9) | 1132 (55·5) | 392 (50·1) | 1329 (54·6) | 1619 (52·2) |
| No | 34,685 (41·7) | 37,424 (40·3) | 1884 (28·5) | 593 (29·1) | 277 (35·4) | 421 (17·3) | 443 (14·3) |
| Don't know | 995 (1·2) | 1,62 (1·4) | 95 (1·4) | 22 (1·1) | 9 (1·2) | 34 (1·4) | 56 (1·8) |
| **¹Dyspnoea 12** | | | | | | | |
| Total Score, median (IQR)* | 9·0 (4, 16) | 8·0 (4, 14) | 9·0 (5, 16) | 8·0 (4, 14) | 7·0 (4, 16·5) | 12·0 (6, 19) | 13·0 (7, 20) |
| Physical Score, median (IQR)* | 7·0 (3, 11) | 6·0 (3, 10) | 7·0 (4, 11) | 6·0 (3, 10) | 6·0 (3, 11) | 8·0 (5, 12) | 9·0 (5, 13) |
| Affective Score, median (IQR)* | 2·0 (0, 5·5) | 1·0 (0, 5) | 2·0 (0, 6) | 2·0 (0, 5) | 1·0 (0, 5) | 3·0 (1, 7) | 4·0 (41, 8) |
| **²PEM Questions (Yes)** | | | | | | | |
| Worsening of fatigue after minimal physical effort* | 13,622 (42·0) | 17,387 (41·6) | 1,21 (51·8) | 560 (49·7) | 218 (55·5) | 1183 (70·3) | 1684 (74·9) |
| Worsening of fatigue after minimal mental effort* | 11,481 (35·2) | 18,111 (43·0) | 1,93 (54·2) | 633 (56·2) | 209 (52·5) | 1141 (67·5) | 1532 (69·7) |
| Exercise makes fatigue symptoms worse* | 13,465 (44·1) | 16,627 (42·4) | 1,89 (52·0) | 528 (49·6) | 177 (50·6) | 1080 (69·1) | 1536 (72·6) |
| **Sleep Quality (0=worse)** | | | | | | | |
| Median (IQR)* | 7·0 (5, 8) | 7·0 (5, 8) | 6·0 (5, 7) | 6·0 (4, 7) | 6·0 (5, 8) | 5·0 (4, 7) | 5·0 (4, 7) |
| **EQ-5D-5L** | | | | | | | |
| EQ5D Visual Analogue Scale, mean (SD)* | 78·4 (18·0) | 78·6 (17·2) | 74·1 (18·7) | 74·2 (18·3) | 75·5 (18·9) | 66·5 (20·4) | 64·7 (21·1) |
| Mobility problems* | 29,751 (27·3) | 24,522 (21·0) | 2113 (28·1) | 643 (27·7) | 236 (26·3) | 1034 (40·5) | 1503 (46·7) |
| Self-care problems* | 9468 (8·7) | 6997 (6·0) | 683 (9·1) | 209 (9·0) | 85 (9·5) | 417 (16·4) | 664 (20·6) |
| Usual activities problems* | 31,546 (28·9) | 29,816 (25·5) | 2843 (37·9) | 860 (37·0) | 305 (34·0) | 1475 (57·8) | 2042 (63·4) |
| Pain/discomfort* | 59,027 (54·1) | 59,494 (50·8) | 4625 (61·6) | 1411 (60·7) | 512 (57·0) | 1833 (71·9) | 2438 (75·7) |
| Anxiety/depression* | 39,704 (36·4) | 46,450 (39·7) | 3908 (52·0) | 1189 (51·2) | 420 (46·8) | 1633 (64·0) | 2108 (65·4) |
| EuroQL-5D Utility Index, mean (SD)* | 0·87 (0·17) | 0·89 (0·14) | 0·84 (0·17) | 0·85 (0·17) | 0·86 (0·17) | 0·78 (0·21) | 0·75 (0·22) |
| **PHQ-9 > = 10*** | 13,538 (13·6) | 16,374 (15·6) | 1697 (25·6) | 515 (25·4) | 159 (20·5) | 932 (43·5) | 1222 (45·9) |
| **GAD-7 > = 10*** | 8815 (8·6) | 11,154 (10·3) | 1120 (16·2) | 345 (16·4) | 109 (13·6) | 584 (25·7) | 732 (26·1) |

Unweighted survey data. Percentages calculated by category after exclusion of missing data. p values show an association of COVID-19 status with outcome. PHQ-9=Patient Health Questionnaire-9. PHQ-9 score was calculated by assigning scores of 0-3 to response categories for 9 questions. PHQ-9 score ≥10: sensitivity 88% / specificity 88% for major depression. GAD-7 = Generalized Anxiety Disorder 7-item scale. GAD-7 score was calculated by assigning scores of 0-3 to response categories for 7 questions. GAD-7 score ≥10: sensitivity 89% / specificity 82% for GAD. 1 Only those reporting shortness of breath asked Dyspnoea-12 questions. Dyspnoea-12 is a patient reported outcome measure (PROM) of 12 questions assessing breathlessness severity. Scores 0-36: higher scores = greater severity of breathlessness; 2 Only those reporting mild or severe fatigue asked PEM questions. EQ5D Visual Analogue Scale is a PROM recording patient's self-rated current health status. Scores 0 (worst possible) to 100 (best possible). EQ-5D-5L is a five-dimension PROM recording a patient's self-rated health state for mobility, self-care, usual activities, pain/discomfort and anxiety/depression. These scores are then mapped to a UK-specific Utility Index anchored at 1 for "perfect health" and 0 for "dead" calculated from reported EQ5D-5L scores across the five dimensions. A χ² test (two-sided) was used to identify differences in proportions across COVID-19 categories. For normally distributed continuous data, analysis of variance (two-way ANOVA) was used to test differences across categories, with Kruskal-Wallis tests used for non-normally distributed data. No adjustments were made for multiple comparisons. *P < 0·0001 (exact P-values: health status P = 0.00003, current symptoms P = 0.0000001, symptoms at infection P = 0.000008, symptom duration P = 0.00009, reduction in activities P = 0.00002, Dyspnoea-12 total P = 0.00003, Dyspnoea-12 physical P = 0.00005, Dyspnoea-12 affective P = 0.00002, PEM 1 P = 0.000007, PEM 2 P = 0.000005, PEM 3 P = 0.000008, sleep P = 0.00009, EQ5D Visual P = 0.000006, mobility P = 0.0000005, self-care P = 0.00000003, activities P = 0.00000001, pain/discomfort P = 0.0000006, anxiety/depression P = 0.0000009, EQ5D Index P = 0.00000001, PHQ-9 P = 0.0000007, GAD-7 P = 0.0000002)

lateral flow devices (LFD) test results, frequency, severity, duration). Current health status included a set of 29 symptoms potentially related to COVID-19 including: (i) loss or change of sense of smell or taste, (ii) coryzal symptoms, (iii) gastrointestinal symptoms, (iv) fatigue-related symptoms, (v) respiratory or cardiac symptoms (vi) memory or cognitive symptoms, (vii) other flu-like and miscellaneous symptoms. Mental and physical health outcomes were collected using the following validated questionnaires:

1. Quality of life/functioning: EuroQol five-dimension five-level (EQ-5D-5L)[31]
2. Assessment of breathlessness in people reporting this symptom: Dyspnoea-12[32]
3. Assessment of post-exertional malaise (PEM) in people reporting fatigue: three PEM-items from the DePaul Symptom Questionnaire[39]
4. Mental health: Generalized Anxiety Disorder 7-item scale (GAD-7)[29] and Patient Health Questionnaire-9 (PHQ-9)[30]

### Data linkage

The UK Health Security Agency (UKHSA) received results of all SARS-CoV-2 PCR tests in England from community settings (Pillar 2)[40]. In addition, members of the public were encouraged to submit results of at-home self-testing using lateral flow devices (Pillar 2)[40]. To obtain additional information on dates of positive SARS-CoV-2 tests, participant study data were linked to their Pillar 2 records using their unique National Health Service (NHS) number and other personal identifiers.

To obtain information on dates of received COVID-19 vaccine doses, participant study data were linked to their NHS records from NHS Digital on COVID-19 vaccination events[41]. This was done using their unique NHS number and other personal identifiers.

### Statistical analyses

In the original REACT programme (REACT-1 and REACT-2) limited information was collected about persistent symptoms as this was not the main objective of these studies. Here, all participants were surveyed about current symptoms (as experienced on day of survey completion) and then later in the questionnaire participants were asked (retrospectively) the date their symptoms they ascribed to COVID-19 initially started and whether they thought their COVID-19 symptoms had resolved (and date of symptom resolution). This information was used to divide the study participants into different categories depending on symptom duration and whether or not they had resolved.

Our primary analyses focused on prevalence of individual symptoms currently reported at the time of questionnaire completion and validated self-reported physical and mental health outcome measures by the following COVID-19 categories of participants:

1. **No COVID**: no history or evidence of SARS-CoV-2 infection or COVID-19;
2. **Asymptomatic or resolved short COVID < 4**: Asymptomatic SARS-CoV-2 infection or COVID-19 symptoms resolved within 4 weeks;
3. **Resolved short COVID ≥ 4 to < 12**: COVID-19 symptoms resolved within 4-12 weeks;
4. **Resolved persistent COVID**: post-COVID-19 symptoms lasting ≥12 weeks but no longer symptomatic; this was further divided into those lasting less than 52 weeks and those lasting ≥52 weeks before resolution.
5. **Ongoing persistent COVID**: post-COVID-19 symptoms lasting ≥12 weeks and ongoing; this was further divided into those lasting less than 52 weeks to date and those lasting ≥52 weeks to date.

We present data on numbers and percent of participants in each of the above categories that are unweighted for potential differential response rates between groups.

We included only symptomatic SARS-CoV-2 infections confirmed by a positive test result (PCR or LFD) in our definition of COVID-19. This included self-reported test positives (survey question), REACT-1 test positives, unvaccinated REACT-2 SARS-CoV-2 IgG test positives and Pillar 2 test positives. Asymptomatic infections were defined as test (PCR or LFD) positives with no reported symptoms. A repeat positive test result was included as a separate infection if performed ≥90 days after a previous positive test[42]. The COVID-19 episode date used was symptom onset date for symptomatic infections or date of a positive

SARS-CoV-2 PCR or LFD test for asymptomatic infections. We excluded individuals with less than 12 weeks follow up from their COVID-19 episode date to their questionnaire completion date.

Index of Multiple Deprivation (IMD) 2019 was used as a measure of relative deprivation, based on seven domains at a community level (Lower Layer Super Output Area, approximately 1500 residents) across England (income, employment, education, health, crime, barriers to housing and services, and living environment)[43]. Participants were allocated to quintiles of deprivation based on their residential postcode. A valid COVID-19 vaccine dose was defined as a date of vaccination 14 days or more prior to the COVID-19 episode date. The Wild-type strain was dominant in the UK prior to December 2020. Alpha dominated between December 2020 and April 2021 followed by Delta (May 2021 to mid-December 2021) and Omicron (late December 2021 onwards)[44].

Continuous variables were presented as median (IQR) or mean (SD), as appropriate. Binary and categorical variables were presented as counts and percentages. A $\chi^2$ test was used to identify differences in proportions across COVID-19 categories. For normally distributed continuous data, analysis of variance (ANOVA F-test) was used to test differences across categories, with Kruskal-Wallis tests used for non-normally distributed data.

We used logistic regression (adjusted Odds Ratios (aOR) and 95% Confidence Intervals (CI) adjusted for age, sex, ethnicity, IMD, comorbidities (presence of a pre-existing health condition) and smoking status (not-current smokers [including ex-smokers] vs. current smokers) to compare current self-reported specific symptoms for those with ongoing persistent symptoms following COVID-19 to those who had never had COVID-19 or had recovered. Further, we used mutually adjusted logistic regression to quantify the associations of age, sex, ethnicity, IMD, comorbidities, smoking status, severity of initial illness, COVID-19 vaccination status, and dominant UK circulating SARS-CoV-2 variant at time of infection with symptoms following COVID-19 lasting ≥12 weeks and lasting ≥52 weeks.

The dataset was converted into a format suitable for survival analysis techniques. Participants were followed up from their COVID-19 symptom onset date until the reported symptom end date (participants provided one date for when all symptoms had resolved) or, if they reported ongoing symptoms, until the survey completion date. We constructed Kaplan-Meier plots of time to symptom end date. To assess factors associated with symptom recovery in participants with symptom persistence beyond 12 weeks, we used an Accelerated Failure Time model[45] with a Log-Normal distribution to quantify the associations between COVID-19 symptom discontinuation beyond 12 weeks and the following factors: age, sex, ethnicity, comorbidities, IMD, smoking status, severity of acute illness, COVID-19 vaccination status, and dominant UK circulating SARS-CoV-2 variant at time of symptom onset. Mutually adjusted Time Ratios (aTR) and 95% Confidence Intervals (CI) were estimated. An aTR >1 is interpreted as a slower symptom recovery rate beyond 12 weeks in participants with COVID-19 symptoms lasting ≥12 weeks.

All tests were two-tailed and p values of less than 0·05 were considered statistically significant. Data were analysed using the statistical package STATA version 15·0.

### Reporting summary

Further information on research design is available in the Nature Portfolio Reporting Summary linked to this article.

## Data availability

The datasets generated or analysed, or both, during the current study are not publicly available because of governance restrictions and the identifiable nature of the data. Requests for access to raw data from the study should be addressed to the corresponding authors and will be answered within 12 weeks. The third party data provided by NHS Digital cannot be made available due to the conditions of the Data

Sharing Agreement between Imperial College London and NHS England. Data from NHS England can be requested directly, see https://digital.nhs.uk/services/data-access-request-service-dars.

## Code availability

STATA code will be made available upon publication to researchers. Requests should be submitted to: react.lc.study@imperial.ac.uk and will be answered within 12 weeks.

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

## Acknowledgements

This work is independent research funded by the National Institute for Health and Care Research (NIHR) and UK Research and Innovation (UKRI): REACT-GE (Genomics England) (UKRI MC_PC_20049) and REACT-LC (Long COVID) (COV-LT-0040). We acknowledge support from the Huo Foundation for this work. This research is part of the Data and Connectivity National Core Study, led by Health Data Research UK in partnership with the Office for National Statistics and funded by UKRI (MC_PC_20029). The REACT-1 and REACT-2 studies were funded by the Department of Health and Social Care in England (DHSC). The views expressed in this publication are those of the authors and not necessarily those of DHSC, NIHR or UKRI. We thank Rob Elliott for his assistance in data acquisition, storage, preparation and governance. We acknowledge the REACT Public Advisory Panel who provided input into all stages of the research. All authors acknowledge Infrastructure support for the Department of Epidemiology and Biostatistics provided by the NIHR Imperial Biomedical Research Centre (BRC). C.A.D. acknowledges support from the MRC Centre for Global Infectious Disease Analysis and the NIHR Health Protection Research Unit (HPRU) in Emerging and Zoonotic Infections (NIHR HPRU award 200907). H.W. is a National Institute for Health Research (NIHR) Senior Investigator and acknowledges support from NIHR Imperial Biomedical Research Centre and the NIHR Applied Research Collaboration North West London. G.S.C. is supported by a National Institute for Health Research (NIHR) Professorship. P.E. is the Director of the MRC Centre for Environment and Health (MR/L01341X/1, MR/S019669/1). P.E. acknowledges support from the NIHR Imperial Biomedical Research Centre and the NIHR Health Protection Research units (HPRUs) in Chemical and Radiation Threats and Hazards and in Environmental Exposures and Health, the British Heart Foundation Centre for Research Excellence at Imperial College London (RE/18/4/34215), Health Data Research UK (HDR UK) and the UK Dementia Research Institute at Imperial (MC_PC_17114).

## Author contributions

C.J.A.: Conceptualization, Formal analysis, Methodology, Writing – Original. Draft, Data Curation, Visualization; B.D: Conceptualization, Methodology, Writing - Original Draft; E.C.: Project administration, Writing - Original Draft; A.L.: Project administration; M.W.: Data Curation, Visualization; A.H.: Supervision, Methodology, Writing - Original Draft; A.A.: Methodology, Writing - Original Draft; C.A.D.: Conceptualization, Methodology, Writing - Original Draft; M.C.: Conceptualization, Methodology; G.S.C: Conceptualization, Funding acquisition, Writing - Original Draft; H.W.: Supervision, Conceptualization, Methodology, Writing - Original Draft; P.E.: Supervision, Conceptualization, Funding acquisition, Methodology, Writing - Original Draft.

## Ethics statement

The study holds ethical approval from South Central-Berkshire B Research Ethics Committee (IRAS IDs: 298404, 259978, 283787 and 298724).

## Competing interests

The authors declare no competing interests.
