## [Peer Review File · Nature Communications]

Long-term health impacts of COVID-19 among 242,712 adults in EnglandREVIEWER COMMENTS

Reviewer #1 (Remarks to the Author):

In this study, the authors sought to assess the factors associated with persistent COVID-19 symptoms, and symptom profiles/quality of life in those who have vs. those who don't have persistent symptoms.

A major strength of the study is a large sample size which is sampled from a general population in the UK. This is important as many prospective studies in this field have relied on opportunistic samples which can be heavily biased and might not provide reliable estimates of prevalences and relative risks.

Below is a list of concerns/queries that I have regarding the manuscript and which I hope will help make it clearer and more relevant to readers:

1) Unfortunately, the recruitment rate and differential non-response mean that the strength of the initial random sampling is somewhat lost. Could the authors use inverse probability weighting to address such selection bias? This could be easily done using the characteristics from Supplementary Table 1, so that estimates are more representative of the English population.

2) I am struggling to understand the timing of the study and this might be a source of significant limitations in the interpretation of findings. Do I understand correctly that:

(i) random samples of the population were repeatedly recruited (in rounds),

(ii) they were surveyed (at the time of recruitment) for retrospective symptoms and time of remission¹¹_{15EP}?

(iii) a subset of them were then followed-up at one specific time point between August and December 2022 at which point they were surveyed once more about their current symptoms?

This would imply that participants reported their symptoms at two time points (one close to their COVID-19 diagnosis and one several months later). If this is so, then how were authors able to state when symptoms ended for individual participants (besides knowing that they ended at some point before the one-time follow-up)? I suspect that participants were specifically asked to retrospectively mention when symptoms ended as alluded to at different parts of the manuscript but this is unclear from the "Study design and participants". If this is the case, then isn't there a risk that recall bias differentially affect people infected at an earlier date (i.e. during the "wild-type" period)?

3) A note should be added in the discussion to make it clear that differences in odds of LC and VLC between time of index infection might have nothing to do with the causative variant and might be due to other factors in the pandemic at those times (since time windows for index infections were wide).

4) Could Figure 1 report number of events in each time window too?

5) Unfortunately, I could not assess Figure 3 as its quality was too poor to read.

6) The use of an accelerated failure time model seems appropriate given the shapes of the curves in Fig. 1. Could the authors provide details on the distribution used in the model? Was this log-logistic?

Reviewer #2 (Remarks to the Author):

-What are the noteworthy results?

Factors associated with persistent COVID-19 symptoms (Female, co-morbidity, wild type variant) and time to recovery (longer for females, comorbidity)

Provides additional information on symptom profile following COVID (and in comparison to those who haven't tested positive).

Corroborate existing findings about worse mental health and quality of life, and demonstrate ongoing concern for a longer time period.

-Will the work be of significance to the field and related fields? How does it compare to the established literature? If the work is not original, please provide relevant references.

Yes I think this work is of significance to the field, due to the larger sample size compared to other studies, the longer duration of follow up and approach to modelling used.

There is however overlap with the following articles which also analysed questionnaire data from participants with and without a positive test for COVID-19. The paper doesn't currently reference either study.

Hastie CE, Lowe DJ, McAuley A, Winter AJ, Mills NL, Black C, Scott JT, O'Donnell CA, Blane DN, Browne S, Ibbotson TR. Outcomes among confirmed cases and a matched comparison group in the Long-COVID in Scotland study. *Nature communications*. 2022 Oct 12;13(1):5663.

Ballering AV, van Zon SK, Olde Hartman TC, Rosmalen JG. Persistence of somatic symptoms after COVID-19 in the Netherlands: an observational cohort study. *The Lancet*. 2022 Aug 6;400(10350):452-61.

In the present study time to recovery was longer for females and people with co-morbidities, and shorter for mixed ethnicity, people living in least deprived areas. These findings corroborate an earlier report from the long COVID in Scotland study which listed female gender, white ethnicity, deprivation and pre-existing health problems with lack of recovery. However, there were differences with the long COVID in Scotland study also reporting severe infection, older age, pre-existing respiratory disease as also being associated with longer time to recover. It would be helpful for the authors to comment on these similarities and differences.

-Does the work support the conclusions and claims, or is additional evidence needed?

The work provided supports the conclusions.

-Are there any flaws in the data analysis, interpretation and conclusions? - Do these prohibit publication or require revision?

There were no obvious flaws, I don't have experience of Accelerated Failure Time model so it may be helpful to gain a statistical perspective.

-Is the methodology sound? Does the work meet the expected standards in your field?

Yes, the methodology is sound and the group have made valuable contributions using similar methods.

-Is there enough detail provided in the methods for the work to be reproduced?

Yes, particularly in combination with the existing papers published on this programme of work.

-Other Major comments

•Overall this was an interesting article with clear methodology and results. The REACT programme of work has been very helpful throughout the COVID pandemic however the introduction seemed to concentrate mostly on REACT which is in some ways useful for framing the work, but actually it would be more useful to provide greater context from other studies which have done similar analyses (principally those mentioned above). The discussion should also make reference to these studies and consider the similarities and differences between the presented work and existing analyses.

-Minor comments

•Figure 2 is difficult to see due to poor resolution. The alignment of the markers (i.e. point estimate and Cis) with the axis labels also makes a reader work hard to interpret. There is no x axis labelling so the magnitude of association isn't clear.

•Figure 3 is also very difficult to make out due to the combination of small text and poor resolution.

•Table 1 the percentage for the total number of individuals with long COVID is incorrect (states 4% but should be 45%)

•Mental health is in the title but not commented on much?

•I wondered why the main characteristics table had been placed in the supplement (Table S2)? For a reader it is nice to have the core information readily available.

•In the tables' smoking status is presented as Yes/No. In the text reference is made to current and non-smokers. It would be helpful to have more detail, how were ex-smokers handled?

•There is an error in the referencing numbers between 9 and 10.

Dr Luke Daines

Reviewer #3 (Remarks to the Author):

Noteworthy results and significance in context: In this paper, the authors used the results of a recent survey sent to a sample of 800 000 individuals included originally in the REACT studies of which 276,840 answered. Among the over 600 000 that were asked to participate without any infection, only 109,186 answered back. The paper mostly reproduces findings that have been described elsewhere regarding risk of Long Covid being higher for the wild type than subsequent variants, the role of preexisting comorbidities in the risk of developing long ongoing symptoms as well as the most reported symptoms after COVID

Delta vs Alpha - Kläser, K., Molteni, E., Graham, M., Canas, L. S., Österdahl, M. F., Antonelli, M., ... & Duncan, E. L. (2022). COVID-19 due to the B. 1.617. 2 (Delta) variant compared to B. 1.1. 7 (Alpha) variant of SARS-CoV-2: a prospective observational cohort study. *Scientific reports*, 12(1), 10904.

Mental health before Long Covid Wang S, Quan L, Chavarro JE, Slopen N, Kubzansky LD, Koenen KC, et al. Associations of Depression, Anxiety, Worry, Perceived Stress, and Loneliness Prior to Infection With Risk of Post-COVID-19 Conditions. *JAMA Psychiatry* [Internet]. 2022 Sep 7 [cited 2022 Sep 26]; Available from: <http://www.ncbi.nlm.nih.gov/pubmed/36069885>

Other comorbidities - Kostev K, Smith L, Koyanagi A, Jacob L. Prevalence of and Factors Associated With Post-Coronavirus Disease 2019 (COVID-19) Condition in the 12 Months After the Diagnosis of COVID-19 in Adults Followed in General Practices in Germany. *Open Forum Infect Dis*

[Internet]. 2022 Jul 4 [cited 2022 Sep 26];9(7). Available from: <https://academic.oup.com/ofid/article/doi/10.1093/ofid/ofac333/6628648>

Symptoms persisting - Ballering A V., van Zon SKR, olde Hartman TC, Rosmalen JGM. Persistence of somatic symptoms after COVID-19 in the Netherlands: an observational cohort study. *The Lancet* [Internet]. 2022 Aug 6 [cited 2023 May 7];400(10350):452–61. Available from: www.thelancet.com

Interestingly, the paper also mentions symptoms experienced by people without COVID symptomatology or with resolved symptoms but don't seem to put it in comparison with studies of symptoms generally reported by the population

Bardel A, Wallander MA, Wallman T, Rosengren A, Johansson S, Eriksson H, et al. Age and sex related self-reported symptoms in a general population across 30 years: Patterns of reporting and secular trend. Reddy H, editor. *PLoS One* [Internet]. 2019 Feb 4 [cited 2022 Apr 23];14(2):e0211532. Available from: <https://dx.plos.org/10.1371/journal.pone.0211532>

In some ways, this paper is interesting in that it brings together all these findings under the same study umbrella to mostly confirm them but there is limited indication of the aims of the study or the hypotheses. With an extending body of literature in the field, one would have expected some more in-depth considerations of the existing work and related works in the field. The introduction is particularly unclear on the aims and scope of the paper and the novelty or absence of novelty of the work

The conclusions appear overall well supported by the work but it would be interesting to understand better the relationship between certain socio-demographic categories and the response rates to disentangle what is a bias in reporting vs what is related to COVID-19. While the authors mention the overall differences with respect to invited population, it is just in passing in the discussion and there is nothing to further compare the reporting across the selected groups. Regarding findings wrt age, some more work may be needed to look at sex/age relationships as some had found some appearing non-linear relationships. Sudre et al Attributes and predictors of Long Covid 2020.

The survey has also the potential to offer more insight on the differences observed in terms of sex, ethnicity, index of deprivation and getting to understand the differences in symptom experiences beyond the categories highlighted in the paper would be of interest.

In addition, understanding the recall bias could be an interesting exercise as well as looking at seasonality effect in addition to main variants as some have found differences in symptom experience according to time of the year

Kifer, D., Bugada, D., Villar-Garcia, J., Gudelj, I., Menni, C., Sudre, C., ... & Lauc, G. (2021). Effects of environmental factors on severity and mortality of COVID-19. *Frontiers in medicine*, 7, 607786.

Overall the methods appear sound and well justified

Response to reviewers (our responses are in red)

Ref.: NCOMMS-23-20257

Long-term physical and mental health impacts of COVID-19 on 242,712 adults in England
Nature Communications

Thank you for sending the helpful comments from the reviewers of our manuscript on the REACT-Long COVID Study. We have addressed these comments in the enclosed response to the reviewers and are pleased to submit our revised manuscript for consideration.

REVIEWER COMMENTS

Reviewer #1:

1) Unfortunately, the recruitment rate and differential non-response mean that the strength of the initial random sampling is somewhat lost. Could the authors use inverse probability weighting to address such selection bias? This could be easily done using the characteristics from Supplementary Table 1, so that estimates are more representative of the English population.

We thank the reviewer for this important question, which is actually not easily addressed. As indicated in the Abstract, Introduction and Methods, the main aim of our analyses is to make comparisons between the population sub-groups (by the COVID-19 categories defined in our methods) rather than making (corrected) estimates of population prevalence. We therefore consider that the model approach used is appropriate rather than the use of weights. Nonetheless, the terms that we have included in the models - for Figures 1-3A and Tables S3-S5 (now Tables S2-S4) – are the same that we would have used for weighting with the exception of Lower Tier Local Authority Area (LTLA), but this would have been covered to some extent by inclusion of Index of Multiple Deprivation in the models.

In fact, production of weights is far from straightforward given the composition of the sample. This would need us to make a number of assumptions and therefore would not necessarily result in more reliable population prevalence estimates, for the reasons listed below:

1. The “recruitment rate” among the REACT-Long COVID participants would need to be worked out separately for each REACT 1 and REACT 2 round and then a composite recruitment rate produced, which itself would need us to make various assumptions about how to combine the different estimates. There were 19 rounds of REACT 1 and 6 rounds of REACT 2 included in REACT-Long COVID, each with different recruitment rates, at different ages.
2. Likewise, we would need to estimate the selection weights for each participant for each REACT 1 or REACT 2 round, before estimating selection weights for the REACT-Long COVID study specifically. These would also need to be worked out separately, as the population definitions were different (REACT 1 was age 5+ years, REACT 2 was age 18+ years), we received periodic updates about population changes from NHS Digital, and also the sampling for REACT 1 switched during the study from aiming for equal numbers per LTLA (and hence over-sampling in more sparse, rural areas) in the earlier rounds to sampling proportional to population in the latter part of the study.
3. In addition, the probability of selection for REACT-Long COVID varied (considerably) according to the sub-groups that made up the sample:
 - a. Had to be a respondent of REACT 1 or 2

- b. Had to give consent to recontact
- c. We first invited all individuals in the following subgroups:
 - i. Individuals from REACT-1 or REACT-2 with a previous history of self-reported test confirmed or suspected COVID-19 who reported persistent symptoms of ≥ 12 weeks (n=52,501)
 - ii. Individuals from REACT-1 who tested positive for SARS-CoV-2 as part of the study (n=13,482)
 - iii. Individuals from REACT-2 who tested positive for SARS-CoV-2 IgG as part of the study and had not been vaccinated at the time (n=85,757)
 - iv. To achieve the 800,000-participant size, a random sample (n=648,260) of all remaining adults not meeting the above criteria was selected.

Thus, the probability of being in the sample was relative to the population, varying recruitment rates across REACT 1 and REACT 2 rounds and relative to whether the participant reported persistent symptoms and tested positive for COVID-19 (note that most suspected COVID-19 cases i.e. no test result, will have been part of last group (n=648,260)).

For all the above reasons we have respectfully decided not to provide weighted estimates but to be clear in text, figures and tables that we are using unweighted estimates. We acknowledge that this requires a caveat on Figure 3B, Table 1 (now Table 2) and Table S2 (now Table 1 in main manuscript) to say that these are observations across the returned unweighted sample. *“Unweighted survey data presented.”* This has been added as a footnote to these items.

We have also added the following text to the methods section “statistical analysis”:

“We present data on numbers and percent of participants in each of the above categories that are unweighted for potential differential response rates between groups.”

2) I am struggling to understand the timing of the study and this might be a source of significant limitations in the interpretation of findings. Do I understand correctly that:

(i) random samples of the population were repeatedly recruited (in rounds)

The REACT 1 and REACT 2 studies sampled random cross-sections of the population in England to quantify community prevalence of virus and of IgG anti-SARS-CoV-2 antibody. There were 19 rounds of REACT-1 and 6 rounds of REACT-2 between May 2020 and March 2022. This is described in the methods section “Study context”.

In the study described in this manuscript (REACT-Long COVID) we obtained a sample of 800,000 adults using as a sample frame all REACT 1 and REACT 2 participants who had consented to both re-contact and data linkage. Therefore, there has been only one round of recruitment into REACT-Long COVID, based on contacting 800,000 of the consented cohort according to the subgroups noted in point (1) 3.c. above and in the methods section “Study design and participants”.

To make this clearer we have replaced the heading “Study context” with “Study sample frame: the REACT programme”. In addition, we have used the name of the study on which the manuscript is based (REACT-Long COVID) in the title and methods section “Study design and participants”.

(ii) they were surveyed (at the time of recruitment) for retrospective symptoms and time of remission

In the original REACT programme (REACT 1 and REACT 2) limited information was collected about persistent symptoms as this was not the main objective of these studies. So, in REACT-Long COVID, all participants were surveyed about current symptoms (as experienced on day of survey completion) and then later in the questionnaire participants were asked the date their symptoms they related to COVID-19 initially started and whether they thought their COVID-19 symptoms had resolved (and date of symptom resolution). This information was used to divide the study participants into the COVID-19 status categories.

We have added a paragraph in the methods section “statistical analysis” to clarify this point.

“In the original REACT programme (REACT-1 and REACT-2) limited information was collected about persistent symptoms as this was not the main objective of these studies. So, in REACT-Long COVID, all participants were surveyed about current symptoms (as experienced on day of survey completion) and then later in the questionnaire participants were asked (retrospectively) the date their symptoms they ascribed to COVID-19 initially started and whether they thought their COVID-19 symptoms had resolved (and date of symptom resolution). This information was used to divide the study participants into different categories depending on symptom duration and whether or not they had resolved.”

(iii) a subset of them were then followed-up at one specific time point between August and December 2022 at which point they were surveyed once more about their current symptoms?

See point (2) ii response above, all information about symptoms, duration, type, date of onset and recovery was collected in REACT-Long COVID between Aug-Dec 2022.

(iv) This would imply that participants reported their symptoms at two time points (one close to their COVID-19 diagnosis and one several months later). If this is so, then how were authors able to state when symptoms ended for individual participants (besides knowing that they ended at some point before the one-time follow-up)? I suspect that participants were specifically asked to retrospectively mention when symptoms ended as alluded to at different parts of the manuscript but this is unclear from the “Study design and participants”. If this is the case, then isn't there a risk that recall bias differentially affect people infected at an earlier date (i.e. during the “wild-type” period)?

Correct, participants were specifically asked to retrospectively mention when symptoms ended. This has been made clearer in the methods section “statistical analysis” (see point (2) ii response to previous comment).

To address the reviewer's point about differential recall, we have added the following text to the discussion:

“There is also a risk that recall bias may have differentially affected participants infected at an earlier date (i.e. during the “Wild-type” period).”

3) A note should be added in the discussion to make it clear that differences in odds of LC and VLC between time of index infection might have nothing to do with the causative variant and might be due to other factors in the pandemic at those times (since time windows for index infections were wide).

We thank the reviewer for this comment. We have added the following text to the section on strengths and limitations in the discussion:

“We can also not rule out residual confounding from other unmeasured time-varying factors, such as behaviour and seasonal weather patterns, for the association between persistent symptoms and dominant variant circulating at the time of infection. However, on balance, studies looking at individuals specifically infected with different SARS-CoV-2 strains support our findings.”

4) Could Figure 1 report number of events in each time window too?

Figure 1 now also reports number of events (number of participants who recovered within previous time window).

5) Unfortunately, I could not assess Figure 3 as its quality was too poor to read.

Figure 3 has been uploaded as a jpeg file as per journal guidance. The authors are also happy to provide the original excel file in which the graphic was produced.

6) The use of an accelerated failure time model seems appropriate given the shapes of the curves in Fig. 1. Could the authors provide details on the distribution used in the model? Was this log-logistic?

Log-Normal distribution was used in the model. This has been added to the methods section “statistical analysis”.

Reviewer #2:

1) There is however overlap with the following articles which also analysed questionnaire data from participants with and without a positive test for COVID-19. The paper doesn't currently reference either study.

We have added the following text to the introduction and cited the two papers suggested by the reviewer:

“Recently, the Long-COVID in Scotland study (Long-CISS), a nationwide study including people with severe, mild and asymptomatic infections and a never infected comparison group, found that 8% of symptomatic participants had not recovered by 6 or 12 months. Participants with previous symptomatic infection were more likely to self-report 24 (of the 26 surveyed) persistent symptoms than people never infected. The largest effect sizes were observed for changes in taste and smell, breathlessness, chest pain, palpitations, and confusion. Similarly, a nationwide population cohort study in the Netherlands (Lifelines COVID-19) with COVID-19-positive cases and matched negative controls concluded that core symptoms of Long COVID were chest pain, difficulties with breathing, lump in throat, pain when breathing, painful muscles, heavy arms or legs, ageusia or anosmia, feeling hot and cold alternately, tingling extremities, and general tiredness.”

2) In the present study time to recovery was longer for females and people with co-morbidities, and shorter for mixed ethnicity, people living in least deprived areas. These findings corroborate an earlier report from the long COVID in Scotland study which listed female gender, white ethnicity, deprivation and pre-existing health problems with lack of recovery. However, there were differences with the long COVID in Scotland study also reporting severe infection, older age, pre-existing respiratory disease as also being associated with longer time to recover. It would be helpful for the authors to comment on these similarities and differences.

We have added the following text to the discussion and cited the paper suggested by the reviewer:

“We found female sex, higher deprivation, having a pre-existing health condition, more severe symptoms at onset, and being infected when the original Wild-type variant was dominant to be associated with having symptoms following COVID-19 persisting for ≥ 12 weeks and ≥ 52 weeks. These variables have previously been identified as risk factors for Long COVID. The risk of persistent symptoms at 12 weeks was slightly lower in the oldest age groups compared to the youngest; this contrasts with other studies and may reflect the higher proportion of older people who are asymptomatic. A comparable population-based study in the UK, the Long-COVID in Scotland study, found older age to be associated with no recovery, with age as a linear predictor in their model.⁷ However, we assumed the relationship between age and persistent symptoms was not linear and included age as a categorical variable in our model as some studies suggest the highest prevalence is found in middle-aged groups.”

And

“Similarly, the Long-COVID in Scotland study found that symptomatic SARS-CoV-2 infection was associated with a wide range of impaired daily activities and reduced health-related quality of life.”

-Other Major comments

3) The REACT programme of work has been very helpful throughout the COVID pandemic however the introduction seemed to concentrate mostly on REACT which is in some ways useful for framing the work, but actually it would be more useful to provide greater context from other studies which have done similar analyses (principally those mentioned above). The discussion should also make reference to these studies and consider the similarities and differences between the presented work and existing analyses.

In addition to the text added to introduction in response Reviewer #2 Point 1, we have provided greater context from other studies through citing a recent meta-analysis in the introduction:

“Estimates of symptom persistence following COVID-19 vary substantially, arguably due to heterogeneous study designs, study settings, follow-up periods and definitions. A recent meta-analysis of 63 studies with data from participants with test-confirmed COVID-19 estimated a pooled symptom prevalence >12 weeks post-infection of 53%. The most common persistent symptoms were fatigue, general pain or discomfort, shortness of breath, cognitive impairment and mental health symptoms. However, most existing studies are based on small sample size, unrepresentative study populations, low response rate or are focused on patients hospitalised with severe COVID-19, so these estimates are unlikely to be representative of the general population.”

We have also added text to the discussion in response to Reviewer #2 Point 2, to consider the similarities and differences in relation to existing studies.

-Minor comments

- Figure 2 is difficult to see due to poor resolution. The alignment of the markers (i.e. point estimate and CIs) with the axis labels also makes a reader work hard to interpret. There is no x axis labelling so the magnitude of association isn't clear.

Figure 2 has been uploaded as a jpeg file as per journal guidance. The authors are also happy to provide the original excel file in which the graphic was produced. Horizontal gridlines and x axis labelling have been added to make the graph easier to interpret.

- Figure 3 is also very difficult to make out due to the combination of small text and poor resolution.

Figure 3 has been uploaded as a jpeg file as per journal guidance. The authors are also happy to provide the original excel file in which the graphic was produced.

- Table 1 the percentage for the total number of individuals with long COVID is incorrect (states 4% but should be 45%)

This has been corrected.

- Mental health is in the title but not commented on much?

The title has been amended to: Long-term health impacts of COVID-19 among 242,712 adults in England: the REACT-Long COVID study.

We have also included a statement regarding the use of the PHQ-9 vs. PHQ-2 questionnaire for measuring depression in Long COVID based on a comment made by a member of the public on our pre-print article.

The following text has been added to the discussion:

“The PHQ-9 scale used is a diagnostic tool for depression. However, some of the somatic questions have been found to be strongly correlated with symptoms that are common in Long COVID, including fatigue, sleep disruption and brain fog. As such, using this scale we might be overestimating the level of depression. This issue was raised by Re’em et al. who suggest the PHQ-2 screening criteria may be more appropriate for Long COVID as they do not include somatic items and simply require a score of 3 or more from the first two questions of PHQ-9. The percentage of participants in our study across all COVID-19 categories with a PHQ-2 score ≥ 3 was lower than the percentage with PHQ-9 ≥ 10 and the difference was more marked in those with ongoing persistent symptoms post-COVID-19 (Supplementary Tables S4 & S5).”

- I wondered why the main characteristics table had been placed in the supplement (Table S2)? For a reader it is nice to have the core information readily available.

Table S2 has been added to the main manuscript as Table 1. The original Table 1 is now referred to as Table 2 in the main manuscript.

- In the tables’ smoking status is presented as Yes/No. In the text reference is made to current and non-smokers. It would be helpful to have more detail, how were ex-smokers handled?

Smoking status has been renamed for clarity “Current smoker” and “Not current smoker”. Ex-smokers were included in the latter category. This is consistent with previous REACT manuscripts. The following text has been added to methods section:

“smoking status (not-current smokers [including ex-smokers] vs. current smokers)”

- There is an error in the referencing numbers between 9 and 10.

This has been corrected.

Reviewer #3:

1. Interestingly, the paper also mentions symptoms experienced by people without COVID symptomatology or with resolved symptoms but don't seem to put it in comparison with studies of symptoms generally reported by the population

We have added some text to the discussion section and cited the paper suggested by the reviewer.

“There have historically been significant secular trends across time regarding self-reported symptom prevalence in general population samples.”

2. In some ways, this paper is interesting in that it brings together all these findings under the same study umbrella to mostly confirm them but there is limited indication of the aims of the study or the hypotheses. With an extending body of literature in the field, one would have expected some more in-depth considerations of the existing work and related works in the field. The introduction is particularly unclear on the aims and scope of the paper and the novelty or absence of novelty of the work

We have expanded the text in the introduction and discussion to consider the wider body of literature. Please see response to Reviewer #2 Point 1, Point 2 and Point 3.

The aims and scope of the study are set out in the last paragraph of the introduction, including some text regarding novelty which we expand on in the discussion.

Introduction text:

“In this study, REACT-Long COVID, we use data from a follow up survey of REACT participants to describe the duration of symptoms in people with a history of symptomatic infection, assess factors associated with symptom persistence beyond 12 weeks (Long COVID) and with recovery after that point. We also compare current self-reported health and quality of life and specific symptoms for those with Long COVID to those who have never had COVID-19 or have recovered. Thus, we tackle some of the limitations of existing studies and strengthen the body of evidence from the few studies which include a negative comparator group.”

Discussion text:

“A strength of our study is that we have addressed some of the limitations of existing studies by having a comparison group and including people in the general population who had severe, mild, and asymptomatic SARS-CoV-2 infections. We compared contemporaneous symptom profiles of community-based adults reporting ongoing persistent symptoms post-COVID-19 versus those who have never had COVID-19 or have recovered. Our study is the largest yet to look at these questions and goes further than previous questionnaire-based studies with COVID-19 negative and never-infected controls by identifying factors associated not only with recovery (yes/no) but rate of recovery, and having multiple comparator groups.”

3. The conclusions appear overall well supported by the work but it would be interesting to understand better the relationship between certain socio-demographic categories and the response rates to disentangle what is a bias in reporting vs what is related to COVID-19. While the authors mention the overall differences with respect to invited population, it is just in passing in the discussion and there is nothing to further compare the reporting across the selected groups.

We have added the following text to the discussion:

“Our questionnaire response rate was 34.6% which is comparable to or higher than similar population-based studies on Long COVID. Questionnaire response rates in the Long-COVID in Scotland Study and a Dutch population-based cohort were 16% and 33-39%, respectively. Like these

studies, our participants were more likely to be female, older, of white ethnicity and from the least deprived areas compared with the general adult population.”

4. Regarding findings wrt age, some more work may be needed to look at sex/age relationships as some had found some appearing non-linear relationships. Sudre et al Attributes and predictors of Long Covid 2020.

We have provided additional text in the discussion highlighting the non-linear relationship we found with age which informed our decision to include age as a categorical variable in our regression models. We have also cited the paper suggested by the reviewer:

“The risk of persistent symptoms at 12 weeks was slightly lower in the oldest age groups compared to the youngest; this contrasts with other studies and may reflect the higher proportion of older people who are asymptomatic. A comparable population-based study in the UK, the Long-COVID in Scotland study, found older age to be associated with no recovery, with age as a linear predictor in their regression model. However, we assumed the relationship between age and persistent symptoms was not linear and included age as a categorical variable in our models as some studies suggest, as did ours, that the highest prevalence is found in middle-aged groups.”

5. The survey has also the potential to offer more insight on the differences observed in terms of sex, ethnicity, index of deprivation and getting to understand the differences in symptom experiences beyond the categories highlighted in the paper would be of interest.

Table 1, now included in the main manuscript, provides a comparison of participant sociodemographics and COVID-19 characteristics by COVID-19 history. This offers the reader more insight into the differences observed in terms of age, sex, ethnicity, and index of deprivation.

Differences in symptom experiences beyond the COVID-19 categories are being explored in more detail from our findings based on semi-structured interviews with REACT-Long COVID participants. Early insights from our interview study can be found here which is now referenced in the paper:

Cooper E, Lound A, Atchison CJ, Whitaker M, Eccles C, Cooke GS, et al. (2023) Awareness and perceptions of Long COVID among people in the REACT programme: Early insights from a pilot interview study. PLoS ONE 18(1): e0280943. <https://doi.org/10.1371/journal.pone.0280943>

6. In addition, understanding the recall bias could be an interesting exercise as well as looking at seasonality effect in addition to main variants as some have found differences in symptom experience according to time of the year.

We have added the following text to the discussion regarding risk of differential recall and cited the paper suggested by the reviewer:

“There is also a risk that recall bias may have differentially affected participants infected at an earlier date (i.e. during the “Wild-type” period). We can also not rule out residual confounding from other unmeasured time-varying factors, such as behaviour and seasonal weather patterns, for the association between persistent symptoms and dominant variant circulating at the time of infection. However, on balance, studies looking at individuals specifically infected with different SARS-CoV-2 strains support our findings.”

REVIEWER COMMENTS

Reviewer #1 (Remarks to the Author):

I thank the authors for providing detailed responses to my comments.

I understand that weighting cannot be easily done and that population prevalence therefore cannot be estimated. This is an unfortunate limitation as it undermines the random sampling. The authors mention that previous studies are based on "unrepresentative study populations" and "low response rate" but it appears that both these issues are also present in REACT-Long COVID.

The completely retrospective assessment of the start and end dates of symptoms (which was unclear to me on first read) is another major limitation. It is really unclear how such a study design can provide reliable estimates of odd ratios between those with and those without COVID-19 as recall bias (differently affecting those with and without COVID-19) is likely to be a major issue.

As a result, it is unclear to me what the main contribution of this study is.

One contribution the study could still have is the effect of variant on COVID-19 consequences. In its current form, this analysis is unable to comment on the effect of variants. This is because people infected at very different times of the pandemic were compared to each other. This is not only confounded by "behaviour and seasonal weather patterns" (as acknowledged by the authors), but also, crucially, by different restrictions put in place during the pandemic (lockdowns, etc) and different knowledge of the effect of SARS-CoV-2 which increases vigilance (e.g. people might be more likely to notice and report ongoing fatigue when it became clear that this is a possible complication of COVID-19). Given the large sample size of the study however, the authors could restrict (for that analysis), the comparison to those infected close to transitions between variants (i.e. on either side of variant transitions). This might not give an answer for rare symptoms (due to lack of statistical power) but would provide relevant information for more common symptoms.

Reviewer #2 (Remarks to the Author):

I'm satisfied that the revised manuscript has addressed the concerns I raised. Thank you.

Reviewer #3 (Remarks to the Author):

The authors are to be commended on the clarity and depth of their answers to the various comments raised during the review.

The added information and development of both introduction and discussion have overall greatly improved the manuscript.

Response to reviewers (our responses are in red), August 2023

Ref.: NCOMMS-23-20257A

Long-term health impacts of COVID-19 among 242,712 adults in England: the REACT-Long COVID study

Thank you for sending the additional comments.

REVIEWER COMMENTS

Reviewer #1:

1. I understand that weighting cannot be easily done and that population prevalence therefore cannot be estimated. This is an unfortunate limitation as it undermines the random sampling. The authors mention that previous studies are based on “unrepresentative study populations” and “low response rate” but it appears that both these issues are also present in REACT-Long COVID.

The completely retrospective assessment of the start and end dates of symptoms (which was unclear to me on first read) is another major limitation. It is really unclear how such a study design can provide reliable estimates of odd ratios between those with and those without COVID-19 as recall bias (differently affecting those with and without COVID-19) is likely to be a major issue.

We thank you for this comment and have sought to clarify throughout that we were not aiming to estimate prevalence of persistent symptoms in the population, but rather to report prevalence in this study. We have amended the Discussion section to emphasise the limitations mentioned. For example, in the abstract we have added “in this study” to the results and removed the phrase “unweighted estimates”. We have also removed the sentence in the introduction which you refer to above (“Thus, we tackle some of the limitations of existing studies and strengthen the body of evidence from the few studies which include a negative comparator group”), and instead address the issue in the discussion.

Relevant sections of the discussion:

“However, in our study population, one in 10 people with symptomatic SARS-CoV-2 infection report symptoms for more than 4 weeks, one in 13 for more than 12 weeks (meeting the WHO definition for “post COVID-19 condition (Long COVID)”⁹), and 1 in 20 for more than 52 weeks (unweighted estimates).”

“A further limitation is that we do not present estimates for population prevalence of persistent symptoms. To do so would require weighting but production of weights is far from straightforward given the composition of our sample. The probability of being in the sample was dependent upon the composition of the base population, varying response rates by sociodemographic group and across REACT 1 and REACT 2 rounds. We also oversampled participants who tested positive for SARS-CoV-2 and who reported persistent symptoms. Producing weights that take account of all these factors would involve making extensive assumptions which would likely introduce unknown biases.”

We have revised the discussion on bias to address the above concern about recall bias:

“We also recognise that the subjective nature of symptoms creates the potential for reporting and recall bias. We used information regarding presence and duration of symptoms rather than whether participants described themselves as having “Long COVID”

to reduce potential reporting bias. The data on symptoms at the time of PCR testing were retrospective which introduces the possibility of recall bias, although we have previously shown that ~~However, previously we have shown that~~ REACT participant reports of symptom onset date ~~produce an epidemic curve that very closely tracks~~ mirrored the epidemic curve.²⁵ There is also a risk that recall bias may have differentially affected reporting of symptoms by participants infected at different times, along with other time-varying factors, such as behaviour, seasonal weather patterns and changing pandemic restrictions, knowledge and expectations,²⁶ which may account for at least part of the association between persistent symptoms and Wild-type infection. However, studies looking at individuals ~~specifically infected~~ with confirmed infections of different SARS-CoV-2 strains also show lower risks with more recent variants.^{27,28} “

2. As a result, it is unclear to me what the main contribution of this study is.

The main contribution and strengths of this study are emphasised in page 11 in the Discussion.

3. One contribution the study could still have is the effect of variant on COVID-19 consequences. In its current form, this analysis is unable to comment on the effect of variants. This is because people infected at very different times of the pandemic were compared to each other. This is not only confounded by “behaviour and seasonal weather patterns” (as acknowledged by the authors), but also, crucially, by different restrictions put in place during the pandemic (lockdowns, etc) and different knowledge of the effect of SARS-CoV-2 which increases vigilance (e.g. people might be more likely to notice and report ongoing fatigue when it became clear that this is a possible complication of COVID-19).

We have added further text to the Discussion section to include the additional factors mentioned (see response to 1 above).

4. Given the large sample size of the study however, the authors could restrict (for that analysis), the comparison to those infected close to transitions between variants (i.e. on either side of variant transitions). This might not give an answer for rare symptoms (due to lack of statistical power) but would provide relevant information for more common symptoms.

We have performed the additional analysis suggested. This is described in Supplementary Table 3 to further explore the effect of variant on persistent COVID-19 symptoms. The results support the findings of our main analysis. However, we acknowledge that studies looking at individuals with confirmed infections with different SARS-CoV-2 strains based on sequencing of positive samples is a more robust method for exploring this association. On balance, studies looking at individuals specifically infected with different SARS-CoV-2 strains support our findings. We have added the following text to the results:

“In a sensitivity analysis to test whether the lower risk with more recent variants was due to unmeasured time-varying factors we restricted the comparison to cases close to the transition period, and also found a reduction in risk for Alpha compared to Wild-type and Omicron compared to Delta (Supplementary Table 3)”.